# Advances in Research on the Regulatory Roles of lncRNAs in Osteoarthritic Cartilage

**DOI:** 10.3390/biom13040580

**Published:** 2023-03-23

**Authors:** Jiaqi Wu, Zhan Zhang, Xun Ma, Xueyong Liu

**Affiliations:** 1Department of Rehabilitation, Shengjing Hospital of China Medical University, No.16, Puhe Street, Shenyang 110134, China; wjq5608@163.com (J.W.); machufei@163.com (X.M.); 2Department of Orthopedics, Shengjing Hospital of China Medical University, Shenyang 110004, China; zhangzhan@sj-hospital.org

**Keywords:** long non-coding RNA, osteoarthritis, cartilage, biomarker

## Abstract

Osteoarthritis (OA) is the most common degenerative bone and joint disease that can lead to disability and severely affect the quality of life of patients. However, its etiology and pathogenesis remain unclear. It is currently believed that articular cartilage lesions are an important marker of the onset and development of osteoarthritis. Long noncoding RNAs (lncRNAs) are a class of multifunctional regulatory RNAs that are involved in various physiological functions. There are many differentially expressed lncRNAs between osteoarthritic and normal cartilage tissues that play multiple roles in the pathogenesis of OA. Here, we reviewed lncRNAs that have been reported to play regulatory roles in the pathological changes associated with osteoarthritic cartilage and their potential as biomarkers and a therapeutic target in OA to further elucidate the pathogenesis of OA and provide insights for the diagnosis and treatment of OA.

## 1. Introduction

Osteoarthritis (OA) is the most common degenerative bone and joint disease. It manifests clinically as pain and limited joint mobility, posing as a major cause of disability and seriously affecting the quality of life of patients. The incidence of OA rises with the aging of the population and improvements in living standards in recent years, affecting about 240 million patients worldwide [1]. However, the etiology of this highly debilitating disease has not been elucidated. Currently, the diagnostic methods for OA are mainly based on symptoms, signs, and imaging staging. However, the severity of symptoms in patients with OA was not completely consistent with the results of imaging studies in clinical practice. During the early stages of OA, a radiologic picture might not reveal any abnormalities [2]. This creates some difficulty in the early diagnosis of OA. Commonly used laboratory tests, such as the erythrocyte sedimentation rate and C-reactive protein tests, mainly play a role in differentiating OA from other types of arthritis (e.g., rheumatoid arthritis). Therefore, we need a simple and feasible means to diagnose OA or determine the development of OA. OA is mainly treated through using non-steroidal anti-inflammatory drugs to relieve joint pain, intra-articular injection of hyaluronic acid or platelet-rich plasma to exert chondroprotective effects, and joint arthroplasty. However, the efficacy of pharmacotherapy in symptomatic relief is far from ideal, while surgical treatment requires more medical resources that impose a heavy financial burden on patients. Therefore, treatment strategies for OA still require further optimization [3,4].

OA has been confirmed to be a degenerative disease that causes pathological changes in the entire joint. It is primarily characterized by articular cartilage degeneration, subchondral bone hyperplasia, synovial inflammation, and osteophyte formation [5,6], among which articular cartilage degeneration is the pathological hallmark underlying the pathogenesis of OA. Therefore, osteoarthritic cartilage has always been a hot topic in OA research [4]. Under normal conditions, the articular cartilage achieves equilibrium between damage and repair. However, the disruption of the dynamic equilibrium during OA results in an imbalance between the mechanisms of damage and repair of cartilage, leading to articular cartilage damage that eventually becomes irreversible. This pathological change involves various biological processes, such as apoptosis and proliferation of chondrocytes, perichondral inflammation, and extracellular matrix (ECM) degradation [7,8,9,10]. With the increasing understanding of OA, the research on the disease has gradually advanced from the histological to the molecular level.

With the rapid development of high-throughput sequencing technology in recent years, differentially expressed long noncoding RNAs (lncRNAs), a kind of multifunctional RNAs longer than 200 nt and between osteoarthritic and normal articular cartilage have been identified in some RNA sequencing (RNA-seq) studies [11]. These lncRNAs were reported to play important roles in the biological processes associated with alterations in osteoarthritic cartilage [12]. An in-depth investigation of the roles of lncRNAs in the progression of OA might lead to major breakthroughs in the diagnosis and treatment of this common disabling disease [13,14]. Therefore, many scholars support the research on the regulatory roles of lncRNAs in osteoarthritic cartilage [15,16], leading to the rapid progress on the research of OA-related lncRNAs. In this review, we summarized the recent progress in this field and discussed the regulatory mechanisms of lncRNAs to further elucidate the pathogenesis of OA with an aim of providing insights for its diagnosis and treatment.

## 2. The Classification and Molecular Mechanism of lncRNAs

LncRNAs were initially discovered as “noise factors” and did not have any function [17]. However, recent studies have shown that lncRNAs account for up to 30% of the genome, can be involved in gene regulation through various pathways, and affect the expression of coding genes [18,19]. LncRNAs can be divided into the following five types according to their distribution at different positions in the genome: (1) sense, (2) antisense, (3) bidirectional, (4) intronic, or (5) intergenic [20] (Figure 1).

LncRNAs are mainly involved in various biological processes, such as epigenetic regulation, transcriptional regulation, and post-transcriptional regulation. Specifically, lncRNAs can mediate different types of histone expression or DNA methylation by recruiting chromatin remodeling or acting as scaffolds for complex modification, which in turn regulates the expression of related genes [21,22] (Figure 2A). LncRNAs could regulates transcription by interfering with RNA pol II recruitment or binding to transcription factors to form complexes [20] (Figure 2B). They could create incomplete complementary duplexes with transcripts of protein-coding genes and by interfering with the cleavage of mRNAs [23] (Figure 2C). They can bind to specific proteins and impact their cellular localization [11,24] (Figure 2D). LncRNAs could also bind to miRNAs through microRNA response elements as a competitive endogenous RNA (ceRNA) or microRNA (miRNA) molecular sponge, thereby affecting miRNA stability and in turn regulating mRNA expression [25] (Figure 2E).

## 3. Regulatory Roles of lncRNAs in Osteoarthritis Cartilage

Because lncRNAs play multiple roles in regulation, it is not surprising that they are involved in various human diseases, such as cancer, cardiovascular disease, diabetes, and arthritis [26]. LncRNAs have been involved in multiple OA stages, such as cartilage destruction, synovial inflammation, subchondral bone invasion, and osteophyte formation [27]. According to current studies on lncRNAs in osteoarthritic cartilage, lncRNAs can modulate mRNA expression mainly by binding to corresponding miRNAs and acting as sponges for miRNAs. This paper summarizes the regulation of lncRNAs in cartilage, specifically, the regulation of lncRNAs on perichondral inflammation, chondrocyte apoptosis, chondrocyte proliferation, and extracellular matrix degradation.

### 3.1. Regulation of Perichondral Inflammation

Currently, the inflammatory microenvironment is a hot topic in the research of OA [28], including the immunomodulation of macrophages in the synovium [29] and monocyte infiltration in the synovium [30], at the core of which are the regulatory roles of inflammatory molecules. The inflammatory response in OA is mainly characterized by many proinflammatory mediators, including cytokines and chemokines, which may promote the production of specific proteins (such as hydrolases) and further degrade the extracellular cartilage matrix, resulting in cartilage tissue destruction [31]. Additionally, changes in the levels of secretion of inflammatory molecules are often accompanied by changes in the expression of certain proteins or the regulation of related cellular pathways in inflammatory models of osteoarthritis [32,33]. Most likely, this is due to lncRNAs modulating the inflammation of osteoarthritic cartilage by regulating microRNAs (miRNAs) or the expression of their downstream targets through competing endogenous RNA (ceRNA) networks. Figure 3 summarizes that the lncRNAs may play regulatory roles in inflammatory responses of osteoarthritic cartilage in recent years.

These studies have suggested the involvement of pvt1 oncogene (PVT1) in the progression of OA. Meng et al. [34] found that PVT1 was upregulated and exerted proinflammatory and proapoptotic effects via the miR-93-5P/HMGB1/TLR4/NF-κB axis in an in vitro LPS-induced human normal chondrocyte injury model (C28/I2). Lu et al. [35] found that the expression of PVT1 was upregulated in the cartilage of patients with OA and IL-1β-treated C28/I2 cells, whereas silencing PVT1 suppressed the inflammatory response and apoptosis, probably via the miR-27b-3p/TRAF3 axis. Similarly, Zhao et al. [36] also observed the high expression of PVT1 in the cartilage of patients with OA and IL-1β-treated chondrocytes, confirming the proinflammatory effects of PVT1. It is therefore possible to delay progress of OA by inhibiting PVT1 expression.

Previous studies have suggested that differentiation antagonizing non-protein coding RNA (DANCR) and HOX transcript antisense RNA (HOTAIR) are associated with the development of OA. It has been identified that they may be involved in the process of osteoarthritis through new pathways in recent years. Zhang et al. [37] found that the expression of DANCR was significantly upregulated in patients with OA. Functional analysis revealed that silencing DANCR inhibited inflammatory responses and chondrocyte proliferation and promoted apoptosis, probably via the regulation of the miR-216a-5p/JAK2/STAT3 signaling pathway. Another study [38] found that DANCR also served as a ceRNA of miR-19a-3p in regulating inflammatory responses, apoptosis, and proliferation of chondrocytes. Wang et al. [39] showed that both the human OA cartilage and the IL-1β-induced OA model had a high level of expression of HOTAIR, which potentially promoted inflammatory responses and apoptosis through the miR-222-3p/ADAM10 axis. Another study [40] uncovered the upregulation of HOTAIR and small glutamine rich tetratricopeptide repeat co-chaperone beta (SGTB) and the downregulation of miR-1277-5p in OA cartilage and LPS-treated CHON-001 chondrocytes. Furthermore, inhibiting HOTAIR suppressed the LPS-induced inflammation and apoptosis, indicating that HOTAIR exerts its regulatory effects by sponging miR-1277-5p and upregulating the expression of SGTB. So, it may be a new strategy to delay progress of OA by inhibiting DANCR and HOTAIR expression.

Small nucleolar RNA host gene family (SNHG) has also become a kind of hot topic lncRNAs in OA research. Yue et al. [41] found that the level of expression of SNHG5 was significantly decreased in IL-1β-treated osteoarthritic tissues and chondrocytes, whereas its overexpression counteracted the proinflammatory effects of IL-1β, probably via the miR-181a-5p/TGFBR3 axis. Similarly, SNHG7 [42], as a ceRNA, was shown to activate the PPARγ pathway by sponging miR-214-5p, thereby inhibiting inflammatory responses and apoptosis. In addition, the study by Lei et al. [43] used an IL-1β-induced inflammatory model of OA to demonstrate that SNHG1 inhibited the expression of proinflammatory cytokines in chondrocytes, probably by activating the miR-16-5p-mediated p38MAPK and NF-κB signaling pathways. Wang et al. [44] also found that SNHG1 inhibited the expression of proinflammatory cytokines in H2O2-treated chondrocytes by targeting the miR-195/IKK-α axis. Contrary to the above findings, Wang et al. [45] found that the expression of SNHG14 was upregulated in OA tissues, whereas downregulation of its expression inhibited apoptosis and the expression of COX-2, iNOS, TNF-α, and IL-6. This was potentially achieved by targeting miR-124-3p to inhibit the FSTL-1-mediated activation of NLRP3 and TLR4/NF-κB signaling pathways. Thus, the specific mechanisms of the SNHG family in the pathogenesis of OA requires further investigation. A number of other studies have reached the exact opposite conclusion. Some of these studies investigated the role of NEAT1 in the inflammatory response in OA. Liu et al. [46] found that the expression of NEAT1 was upregulated in OA cartilage tissues and chondrocytes, whereas downregulation of its expression inhibited inflammatory responses and apoptosis of chondrocytes and decreased the protein levels of MMP-3, MMP-13, and ADAMTS-5. Another study by Wang et al. [47] revealed that the expression of NEAT1 was downregulated in OA tissues, whereas downregulation of its expression increased the rate of apoptosis and the levels of inflammatory cytokines released from OA chondrocytes (OACs). Other studies explored the role of XIST in the inflammatory response in OA. A study by Li et al. [48] on the inflammatory microenvironment of joints revealed a significant increase in the levels of expression of XIST in OA tissues. Further, knockdown of XIST improved the inflammatory microenvironment in OA by acting on M1 macrophages, subsequently affecting the apoptosis of cocultured chondrocytes. In contrast, Lian et al. [49] found that the expression of XIST was downregulated in OA tissues and a cellular model of OA, whereas upregulation of its expression enhanced the viability and inhibited apoptosis and IL-1β-induced inflammatory responses in CHON-001 and ATDC5 cells.

In addition to the aforementioned lncRNAs, there are a relatively small number of studies for other lncRNAs in recent years. Activated by TGF-beta (ATB) [50], whose expression was downregulated in LPS-injured cells, is a ceRNA that inhibits the MyD88/NF-κB and p38 MAPK signaling pathways by sponging miR-233, thereby exerting a protective effect against LPS-induced inflammatory injury. Moreover, HULC [51] suppressed the inflammatory responses and apoptosis via the miR-101/NF-κB/MAPK axis, while MSC-AS1 [52] regulated the JAK2/STAT3 signaling pathway by sponging miR-369-3p, thereby exhibiting anti-inflammatory and antiapoptotic effects. Xie et al. [53] found that MEG8, whose expression was downregulated in IL-1β-treated C28/I2 cells, exerted a protective effect against IL-1β-induced inflammatory responses and apoptosis of chondrocytes by regulating the PI3K/AKT signaling pathway. Furthermore, the expression of HOTAIRM1-1, unlike HOTAIR, was downregulated in tissues of patients with OA. HOTAIRM1-1 suppressed inflammation and inhibited cell proliferation by interacting with miR-125b [54]. HOTAIRM1-1 was also reported [55] to exert an antiapoptotic effect through the miR-125b/BMPR2/JNK/MAPK/ERK axis. It is also necessary to investigate further about the specific mechanisms that these lncRNAs contribute to the pathogenesis of OA.

In addition, cancer susceptibility 2 (CASC2) [56] and cancer susceptibility 19 (CASC19) [57] have been demonstrated to exhibit proinflammatory and proapoptotic effects by regulating IL-17 and the miR-152-3p/DDX6 axis, respectively. Furthermore, Zhang et al. [58] found that the levels of ADP ribosylation factor related protein 1 (ARFRP1) and toll like receptor 4 (TLR4) were increased, whereas those of miR-15a-5p were decreased in lipopolysaccharide (LPS)-treated chondrocytes, and that ARFRP1 promoted the expression of inflammatory molecules, such as IL-6 and TNF-α, by regulating the NF-κB pathway. Luo et al. [59] uncovered that the expression of MFI2 Antisense RNA 1 (MFI2-AS1) was increased in OA tissues and LPS-treated C28/I2 cells, whereas the silencing of MFI2-AS1 alleviated the inflammatory response and apoptosis, probably by regulating the miR-130a-3p/TCF4 axis. In a model of LPS-induced OA cell injury, Liu et al. [60] showed that the overexpression of TNF- and HNRNPL-related immunoregulatory long non-coding RNA (THRIL) in the LPS-induced cell injury model of OA exacerbated the inflammatory injury of cells by downregulating the expression of miR-125b to enhance the LPS-induced activation of the JAK1/STAT3 and NF-κB pathways. Inhibition of the expression of THRIL had the opposite effects, suggesting that THRIL exerts proinflammatory effects. More interestingly, the expression of the aforementioned proinflammatory lncRNAs was found to be upregulated in OA tissues and inflammatory cell models. However, Wang et al. [61] showed that the expression of THUMP domain containing 3 antisense RNA 1 (THUMPD3-AS1) was downregulated in OA cartilage tissues and IL-1β-stimulated chondrocytes, despite finding that its overexpression promoted inflammatory responses and inhibited apoptosis. 

Long intergenic noncoding RNAs (lincRNAs) have also been involved in the onset and development of OA. In particular, LINC00265, LINC00461, LINC00473, LINC01534, and LINC02288 [62,63,64,65,66] have been demonstrated to promote inflammatory responses by sponging related miRNAs. Among these lincRNAs, LINC00265, LINC00473, and LINC02288 have also been shown to promote apoptosis. Moreover, Pearson MJ et al. [67] found the lincRNA p50-associated cyclooxygenase 2–extragenic RNA (PACER) and 2 chondrocyte inflammation–associated lincRNAs (CILinc01 and CILinc02) were down-regulated in hip and knee OA cartilage. Among them, PACER and CILinc01 were associated with the IL-1 inflammatory response in chondrocytes, and CILinc02 was associated with IL-7.

### 3.2. Regulation of Chondrocyte Apoptosis 

Apoptosis, also known as programmed cell death, is a programmed process of cell self-destruction triggered in incapacitated or damaged cells by activating apoptotic pathways or downstream factors [68]. Chondrocyte apoptosis is an essential factor in OA development. The mechanisms of chondrocyte apoptosis in OA are complex, involving several reported key pathways, such as nitric oxide, Bax/Bcl-2, NF-κB, and cysteine-aspartic protease (caspase)-related pathways [69,70,71,72]. In recent years, studies on the pathogenesis of OA at the molecular level have identified many lncRNAs that regulate chondrocyte apoptosis in various ways. The specific regulatory pathways or factors are summarized in Figure 4.

The earliest identified lncRNAs, XIST and H19, can influence OA processes through multiple pathways. In addition to modulating inflammatory responses, Liu et al. [73] found that XIST was highly expressed in OA cartilage tissues and IL-1β-treated chondrocytes, whereas knocking out XIST enhanced cell viability and inhibited apoptosis and protein degradation in ECM, indicating that XIST might exert its regulatory effects through the miR-149-5p/DNMT3A axis. However, Li et al. [74] showed that the expression of XIST was significantly increased in cartilage samples from patients with OA, whereas knockdown of XIST significantly abolished the inhibitory effect of IL-1β on the proliferation of OACs and promoted the IL-1β-induced apoptosis, which was associated with the miR-211/CXCR4 axis. Zhang et al. [75] uncovered that both OA samples and IL-1β-treated chondrocytes had significantly upregulated expression of H19. Moreover, its overexpression inhibited the proliferation and induced apoptosis in OACs, probably by sponging miR-106a-5p. Yang et al. [76] also observed the upregulation of H19 in clinical samples of OA cartilage tissues. Silencing H19 not only reduced the level of apoptosis but also promoted the proliferation of chondrocytes, potentially via the H19/miR-140-5p regulatory axis. Therefore, it is possible to ameliorate the severity of OA by inhibiting the expression of H19 or increasing the expression of XIST.

HOTAIR and growth arrest specific 5 (GAS5) were reported to regulate chondrocyte apoptosis. There are some novel findings about them. He et al. [77] showed that the abnormally high level of expression of HOTAIR resulted in the miR-130a-3p-mediated inhibition of chondrocyte autophagy, leading to the massive apoptosis of chondrocytes and the upregulation of the expression of apoptotic genes, eventually causing OA. Hu et al. [78] found that the expression of HOTAIR was upregulated in OA cartilage tissues, and subsequent functional studies revealed that HOTAIR exacerbated ECM degradation and chondrocyte apoptosis, which might have been associated with the miR-17-5p/FUT2-mediated activation of the Wnt/β-catenin pathway. Chen et al. [79] found that both the articular cartilage tissue of mice with OA and IL-1β-treated chondrocytes had significantly upregulated expression of HOTAIR. Moreover, overexpression of HOTAIR was shown to inhibit the IL-1β-induced proliferation of chondrocytes and promote apoptosis and ECM degradation. These effects were reversed after knocking down HOTAIR. Taken together, these findings demonstrated that HOTAIR might be involved in the progression of OA by targeting and regulating the miR-20b/PTEN axis. In a study by Ji et al. [80], silencing GAS5 promoted the proliferation and inhibited the apoptosis of OACs and triggered the G1-phase cell cycle arrest, which might be attributed to the miR-34a/Bcl-2 axis-mediated regulatory effect of GAS5 on the biological behaviors of chondrocytes. Gao et al. [81] concluded that the expression of GAS5 was upregulated in the serum and cartilage tissues of patients with KOA, whereas its downregulation led to the inhibition of chondrocyte apoptosis via the downregulation of miR-137. In conclusion, these studies have revealed that HOTAIR and GAS5 may also be potential therapeutic targets in OA.

The SNHG family is also an important component of chondrocyte apoptosis regulation. SNHG5, SNHG7, and SNHG15 exert antiapoptotic effects via different pathways, despite their low levels of expression in OACs and OA tissues. Jiang et al. [82] found that the expression of SNHG5 was downregulated in OA cartilage tissues and its knockdown enhanced the IL-1β-induced apoptosis in chondrocytes. It is possible that the antiapoptotic effect of SNHG5 was achieved by sponging miR-10a-5p to regulate the expression of H3F3B. Similarly, a functional study by Tian et al. [83] showed that upregulation of SNHG7 promoted the proliferation, and inhibited the apoptosis and autophagy, of OA cells. This suggested that SNHG7 might modulate the progression of OA by sponging miR-34a-5p and regulating the expression of synoviolin 1 (SYVN1). Zhang et al. [84] also observed the downregulation of SNHG15 in OA cartilage tissues and IL-1β-treated chondrocytes. Downregulation of SNHG15 might improve the viability of chondrocytes and reduce the level of chondrocyte apoptosis and ECM degradation via the miR-141-3p/BCL2L13 axis. The above-mentioned studies were published in 2020, suggesting the importance of the SNHG family in OA research at the molecular level in recent years.

Besides the above-mentioned lncRNAs, some other lncRNAs have received significant attention in recent years. Zinc finger NFX1-type containing 1 antisense RNA 1 (ZFAS1) is a lncRNA that inhibits apoptosis. Li et al. [85] found low levels of expression of ZFAS1 in OA tissues, and further analysis showed that ZFAS1 accelerated the proliferation and inhibited the apoptosis of chondrocytes by regulating the miR-302d-3p/SMAD2 axis. Similarly, another group [86] reported that OACs had a lower level of expression of ZFAS1 than normal chondrocytes, and overexpression of ZFAS1 enhanced the viability, proliferative capacity, and migratory capacity of OACs whilst inhibiting the apoptosis and ECM synthesis of chondrocytes by targeting the Wnt3a signaling pathway. Han et al. [87] found that the expression of TUG1 was significantly upregulated in OA tissues and OA tissue-derived chondrocytes, as well as in IL-1β-treated C28/I2 cells. Further, knocking down TUG1 promoted the proliferation and inhibited apoptosis and ECM degradation, potentially via the miR-320c/FUT4 axis-mediated regulatory effect of TUG1. Similarly, Li et al. [88] observed elevated levels of expression of TUG1 in OA tissues and IL-1β-treated chondrocytes, whereas knocking down TUG1 enhanced cell viability and inhibited apoptosis through the miR-17-5p-mediated downregulation of the expression of fucosylransferase (FUT1). Jiang et al. [89] found high levels of expression of lincRNA-Cox2 in the OA cartilage tissue of a mouse model and IL-1β-treated chondrocytes. In contrast, knockout of lincRNA-Cox2 promoted the proliferation and inhibited the apoptosis of chondrocytes. This might be attributed to its ceRNA sponge effect against miR-150 and its regulatory effect on the Wnt/β-catenin pathway. LINC00511 is another possible sponge of miR-150. Zhang et al. [90] uncovered that knockout of LINC00511 inhibited the apoptosis and promoted the proliferation and ECM synthesis in chondrocytes, indicating that LINC00511 might exert its effect on SP1 by sponging miR-150-5p. A recent study by Tang et al. [91] found that upregulation of PILA regulated the NF-κB pathway by increasing the level of expression of TAK1 via PRMT1/DHX9, thereby promoting chondrocyte apoptosis. 

The following studies have reached to opposite conclusions. In recent years, a number of studies have observed the low levels of expression of MEG3 in OA tissues or simulated OACs. Huang et al. [92] found that treatment of chondrocytes with IL-1β downregulated the expression of MEG3, whose overexpression inhibited inflammatory responses and apoptosis. Further studies revealed that the above-mentioned observation might be attributed to MEG3, which induced the expression of Krüppel-like factor 4 (KLF4) via sponging miR-9-5p. Chen et al. [93] found that the overexpression of MEG3 induced the proliferation of IL-1β-treated chondrocytes and inhibited their apoptosis and ECM degradation, which might be related to the target of the lncRNA MEG3, that is, the miR-93/TGFBR2 axis. Wang et al. [94] also observed that MEG3 regulated the expression of forkhead box O1 (FOXO1) by acting as a ceRNA of miR-361-5p, thereby promoting the proliferation of OACs and inhibiting their apoptosis and ECM degradation. It is interesting to note that Xu et al. [95] also showed that the expression of MEG3 was downregulated in a rat model of IL-1β-induced OA, but the results of their functional analysis were contradictory to those of the previous study, suggesting that the antiproliferative and proapoptotic effects of MEGs were achieved through the regulation of the miR-16/SMAD7 axis. In 2021, Shi et al. [96] reported an experiment carried out on minichromosome maintenance complex component 3 associated protein antisense RNA 1 (MCM3AP-AS1) in a simulated OA inflammatory environment using the chondrocyte cell line CHON-001 and IL-1β-treated ATDC5 cells. They showed that the level of expression of MCM3AP-AS1 was decreased in OA cartilage tissues, whereas upregulation of its expression enhanced the viability and migratory capacity of CHON-001 and ATDC5 cells and inhibited their apoptosis and inflammatory responses. In 2022, Xu et al. [97] found that the expression of MCM3AP-AS1 was upregulated in patients with OA and IL-1β-treated chondrocytes. Moreover, MCM3AP-AS1 promoted the apoptosis and ECM degradation of chondrocytes by regulating the miR-149-5p/Notch1 axis, which was consistent with the findings of an earlier study by Gao et al. [98] in patients with OA and healthy controls. To conclude, the above-mentioned studies on GAS5, DANCR, and NEAT1 either reached an opposite conclusion or obtained different levels of expression in tissues [99,100,101,102]. This might be due to differences in the levels of expression of noncoding RNAs in different experimental subjects.

### 3.3. Regulation of Chondrocyte Proliferation and ECM Degradation 

Chondrocytes are the only cells found in cartilage. They are distributed in the extracellular matrix of cartilage. Current research has found that chondrocytes play an important role in maintaining the stability of the extracellular matrix. Furthermore, components of the extracellular matrix are essential for cell proliferation, differentiation, and migration [103,104]. The internal environment of cartilage cells is stable under normal circumstances. However, this stability is broken in OA due to increased apoptosis or decreased cell proliferation, leading to degradation of the extracellular matrix of cartilage cells, a decrease of proteoglycans in the matrix, and the change in concentration, composition ratio, and molecular chain of various frame proteins [105,106]. Therefore, any factor affecting cell proliferation or ECM degradation might also influence the onset and development of OA to some extent. Recent studies have shown that lncRNAs can affect the viability and proliferation of cells, as well as ECM degradation, by regulating miRNAs or downstream pathways and proteins (Figure 5 and Figure 6). Thus, the lncRNAs affecting both chondrocyte proliferation and extracellular matrix degradation will be discussed in this review.

Metastasis associated lung adenocarcinoma transcript 1 (MALAT1) is the most representative lncRNA capable of regulating cell proliferation and ECM degradation. It has previously garnered great attention in cardiovascular disease and cancer research [107,108]. It is highly expressed in endothelial cell nuclei and has been strongly associated with the proliferation of endothelial cells. In recent years, numerous studies on lncRNAs and OA have focused on MALAT1. Liang et al. [109] found that the tissues of patients with OA had an increased level of expression of MALAT1. Conversely, subsequent knockdown of its expression via transfection with small interfering RNA significantly suppressed the proliferation of human OACs, suggesting that MALAT1 exerts its function via the activation of the OPN/PI3K/Akt pathway by sponging miR-127-5p. In vitro simulation of OA by Zhang et al. [110] also showed the upregulation of MALAT1 in IL-1β-treated chondrocytes. In contrast, knockdown of MALAT1 inhibited the proliferation of IL-1β-treated chondrocytes and promoted their apoptosis. In addition, they also found that MALAT1 knockdown reduced the degree of ECM degradation. These processes were assumed to be regulated by MALAT1 via the miR-150-5p/AKT3 axis. Interestingly, Liu et al. [111] also found that the expression of MALAT1 was upregulated in OA samples and IL-1β-treated chondrocytes, but their functional analyses reached the opposite conclusion to that of Zhang et al., suggesting that MALAT1 inhibited the viability of IL-1β-induced chondrocytes and promoted ECM degradation in the cartilage. Inhibiting the expression of MALAT1 may thus be a possible strategy to prevent the further development of OA.

KCNQ1 opposite strand/antisense transcript 1 (KCNQ1OT1) and HOXA distal transcript antisense RNA (HOTTIP) have also become a focus of research in recent years. Wang et al. [112] found that the increased level of expression of KCNQ1OT1 enhanced the viability and migratory capacity of CHON-001 cells and inhibited ECM degradation. It is possible that overexpression of KCNQ1OT1 exerted these effects by sponging miR-126-5p to target transcriptional repressor GATA binding 1 (TRPS1). Liu et al. [113] uncovered that OACs had a lower level of expression of KCNQ1OT1 than normal chondrocytes. Conversely, upregulation of KCNQ1OT1 significantly enhanced the viability of OACs, reduced the release of inflammatory cytokines and matrix metalloproteases, and inhibited apoptosis by sponging miR-218-5p to activate the PI3K/AKT/mTOR pathway. Using a chondrogenesis model based on human mesenchymal stem cells (hMSCs) and an in vivo animal model, Mao et al. [114] found that the expression of the lncRNA HOTTIP was significantly upregulated in OA cartilage tissues and that knocking down HOTTIP promoted chondrocyte proliferation, potentially through the ceRNA regulatory network of HOTTIP/miR-455-3p/CCL3. In contrast, He et al. [115] reached the opposite conclusion. Despite observing the upregulation of HOTTIP in OA cartilage tissues, subsequent knockdown of its gene suppressed the proliferation of OA cartilage cells and increased their levels of apoptosis, whereas overexpression of HOTTIP had the opposite effect. The latest studies by Xu et al. [116] and Qian et al. [117] suggested that LINC00707 inhibited chondrocyte proliferation and enhanced ECM degradation. Xu et al. found that the level of expression of miR-199-3p decreased with the increased expression of LINC00707 in OA tissues. Although the interrelationship of the levels of expression between these two RNAs has not yet been clarified, it was suggested that LINC00707 inhibited the proliferation of OACs by sponging miR-199-3p. Qian et al. also observed an increased level of expression of LINC00707 in OA tissues and subsequent experimental validation showed that ECM degradation was inhibited following the silencing of LINC00707. However, they found that LINC00707 exerted the above-mentioned effects by serving as a ceRNA for miR-330-3p to regulate the expression of follicle-stimulating hormone receptor (FSHR). In addition to modulating inflammatory responses and apoptosis, some of the aforementioned lncRNAs have also been studied individually for their effects on cell proliferation and ECM degradation. Yao et al. [118] found that PVT1, whose expression was upregulated in human chondrocytes, acted as an endogenous RNA sponge to suppress the expression of miR-140, thereby promoting the expression of MMP-13 and ADAMT-5 and ultimately leading to an increased degree of ECM degradation. Wang et al. [119] suggested that XIST acts as a ceRNA of miR-1277-5 to affect ECM degradation and further verified that downregulation of XIST reduced the degree of ECM degradation in a rat model of OA. Using exosome transfection in cultured chondrocytes, Tan et al. [120] observed that H19 promoted the proliferation and migration of chondrocytes and inhibited ECM degradation by targeting the miR-106b-5p/TIMP2 axis. Shen et al. [121] found that OA tissues had a significantly downregulated expression of SNHG5, and subsequent MTT and transwell assays confirmed that SNHG5 promoted chondrocyte proliferation and migration by acting as a ceRNA that sponges miR-26a to regulate the expression of SOX2. Tang et al. [122] observed that the cartilage of patients with OA had higher levels of expression of TUG1 and MMP-13 than those of normal individuals. Subsequent transfection of OACs with a plasmid expressing TUG1 confirmed that TUG1 regulated ECM degradation via the TUG1/miR-195/MMP-13 axis. These results suggested that these lncRNAs might be a potential therapeutic target for OA.

With the exception of the aforementioned lncRNAs, linc-regulator of reprogramming (ROR) [123] and actin filament associated protein 1 antisense RNA 1(AFAP1-AS1) [124] were also shown to promote chondrocyte proliferation, while LINC00623 [125] was reported to reduce the degree of ECM degradation. Besides these, other lncRNAs, such as prostate androgen-regulated transcript 1(PART1) [126], FOXD2 adjacent opposite strand RNA 1(FOXD2-AS1) [127], and PCGEM1 prostate-specific transcript (PCGEM1) [128], are known to affect both chondrocyte proliferation and ECM degradation. However, only a few studies suggest their potential existing role in the proliferation of OA chondrocytes or the degradation of extracellular matrix. In fact, we noticed that, given the mechanistic complexity of lncRNAs, most of the above studies were carried out on lncRNAs in osteoarthritic cartilage with different pathological changes and rarely focused on a particular pathological change. Therefore, to better represent the expression and regulation of lncRNAs in OA, we have comprehensively summarized all lncRNAs described in this review in Table 1 (animal data) and Table 2 (human data).

## 4. New Perspectives and Limitations of lncRNAs in the Diagnosis and Treatment of OA

The identification of the lncRNA molecules and the investigation of its action mechanism provide new insights into the occurrence and development of diseases, in addition to the discovery of new approaches for the diagnosis and treatment of diseases. LncRNA molecules had hitherto been considered biomarkers for diagnostic and therapeutic targets in cancer, pulmonary hypertension, autoimmune diseases, and other diseases [129,130,131]. After an in-depth study of lncRNAs in the pathogenesis of osteoarthritis, researchers discovered that the expression of lncRNAs differs between OA patients and healthy individuals. Therefore, these variations could serve as biomarkers to aid in the diagnosis of OA.

In a recent study, Jiang et al. [132] screened lncRNAs differentially expressed in the serum by RNA-seq and validated them by tissue qRT-PCR and bioinformatics analysis. They found that Linc00617 could effectively distinguish healthy individuals from patients with OA and that this lncRNA was not associated with OA duration; thus, Linc00617 may be a biomarker for the early diagnosis of OA. Similarly, Dang et al. [133] analyzed blood samples from 98 patients with OA and healthy individuals and found that the expression of ATB was significantly downregulated in the serum of patients with OA. The team concluded that this index could distinguish patients with OA from healthy people; thus, ATB in the serum could be used as a reliable diagnostic marker for OA. In another study, in which peripheral blood samples were collected from 130 patients with OA and 100 healthy individuals [134], it was found that lncRNA H19 expression was increased in the peripheral blood of patients with OA, suggesting that H19 has an excellent diagnostic value for OA. In addition to blood samples, Zhao et al. [135] found that lncRNA PCGEM1 was effective in distinguishing patients with OA from healthy individuals in synovial fluid samples and that the more severe the degree of OA, the higher the PCGEM1 expression. This finding also suggests that PCGEM1 may serve as a biomarker for diagnosing OA and assessing its severity. These latest studies suggest a potential diagnostic role for lncRNAs. These recent studies all suggest a potential diagnostic role for lncRNAs, but the sample sizes included in these studies are small. The value of the aforementioned lncRNA molecules as diagnostic indicators of OA will increase if multicenter studies with greater sample sizes are conducted.

Unfortunately, osteoarthritis is primarily treated with non-steroidal anti-inflammatory medications and surgery [136]. Although a great number of lncRNA molecules express distinctively in normal or osteoarthritic cartilage and have been discovered to affect the pathophysiological process of osteoarthritis cartilage through corresponding target genes or downstream pathways, there is currently no lncRNA molecule that can be directly applied in clinical treatment. The OA clinical trial study presently registered in the clinical trials registries (clincialtrials.gov) is mainly focused on examining the postoperative joint microenvironment and the molecules and cells that are essential to the osseointegration process of joint prosthetics. 

The challenges of lncRNA and other molecular targeting therapies mainly lie in targeting and specificity, which are predominately manifested as the adverse targeting effect induced by silencing molecules in non-target cells, or the miss-target effect caused by beneficial molecules failing to achieve the preset goals [137,138]. At present, RNA interference (RNAi), which hinders lncRNAs expression by small interfering RNA (siRNA) [139], is one of the potentially useful techniques for the targeting of lncRNAs. The CRISPR/Cas-9 system controls lncRNAs expression and reduces theirs level [140]. Antisense oligonucleotides technology controls lncRNAs expression by inducing the binding of silent RNase H to its lncRNA molecules [141] and inhibits or stabilizes the expression of lncRNAs by binding several small molecules to lncRNAs [142,143]. In the study of specific molecular delivery, it is possible to transport target lncRNAs through nanoparticles and extracellular vesicles modified [144,145] and plasmid [146] as carriers. However, there are limited studies on these methods in OA. Therefore, more experimental support is required to research the approaches for the clinical application of the therapeutic potential of lncRNA molecules in OA.

## 5. Summary and Outlook

The pathogenesis of OA has not yet been fully elucidated, as it is a multifactorial process. Previous studies of our research team mainly focused on protein-coding genes in OA [147,148]; however, we found that recent studies on its pathogenesis have shifted focus to the roles of noncoding RNAs by literature search. Therefore, we summarized the lncRNAs that play a role in osteoarthritic cartilage in this review. It is now believed that there are some important cross-talks between cartilage and subchondral bone, synovial tissue fibroblasts, and chondrocytes in the development of OA, so it is possible that some lncRNAs that play a role in OA cartilage pathogenesis also regulate other pathological processes in OA. For example, some investigators found that PCGEM1 [149] and MEG3 [150] could regulate the proliferation of synoviocytes to affect the development of OA, as mentioned in this article. Regrettably, there are few studies on lncRNAs in OA subchondral bone. Tuerlings M et al. [151] identified some differentially expressed lncRNAs by sequencing in OA subchondral bone, but further experiments are needed to verify the functions of these lncRNAs. Accordingly, lncRNAs that play roles in crosstalk between cartilage and other joint structures might be a future research direction.

There are still many limitations in current studies on lncRNAs. For instance, most researchers employed in vitro cell-based assays and animal models, but the applicability of their findings to humans still requires further investigation and validation. Therefore, the idea of using a certain LncRNA molecule as a diagnostic biomarker or as a target for drug therapy still requires extensive basic experiments to determine a relatively reliable signal pathway, and thus develop relevant reagents or drugs. In conclusion, lncRNAs are a class of molecules with great clinical application prospects and have the potential to serve as diagnostic and therapeutic biomarkers for OA, but there are still many difficulties to overcome.

## Figures and Tables

**Figure 1 biomolecules-13-00580-f001:**
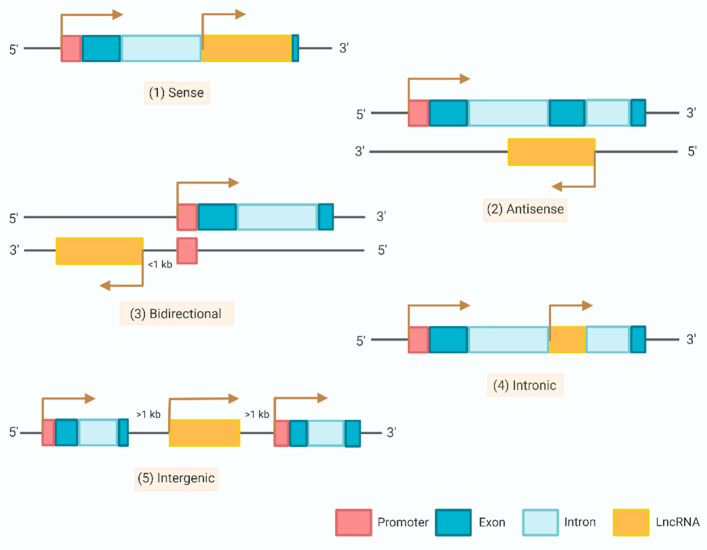
Depending on their genomic location relative to nearby protein-coding genes, lncRNAs could be classified as sense (lncRNAs that transcribe in the same di-rection that overlaps coding exons), antisense (lncRNAs that transcribe in the opposite direction that overlaps coding exons), bidirectional (lncRNAs with identical promoters to coding genes but transcript oppositely), intronic lncRNAs (lncRNAs that located between two introns), or intergenic lncRNAs (lncRNAs that located between two coding genes). This figure was created with Biorender.com (accessed on 6 October 2022).

**Figure 2 biomolecules-13-00580-f002:**
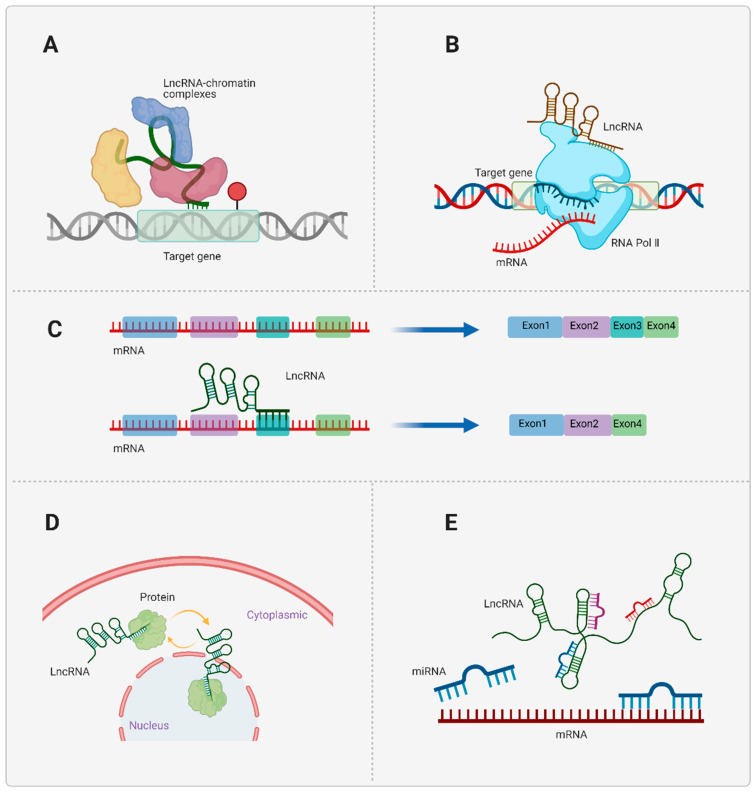
The molecular mechanisms of lncRNAs. (**A**) Participate in epigenetic regulation by recruiting chromatin. (**B**) Regulation of transcription by interfering with recruitment of RNA polymerase II. (**C**) Interfere process of mRNA splicing through binding with complementary sequences. (**D**) Bind to specific proteins and influences their cellular location. (**E**) Serve as a molecular sponge to bind miRNA to form a lncRNA–miRNA axis that in turn modulate mRNA expression. This figure was created with Biorender.com (accessed on 4 October 2022).

**Figure 3 biomolecules-13-00580-f003:**
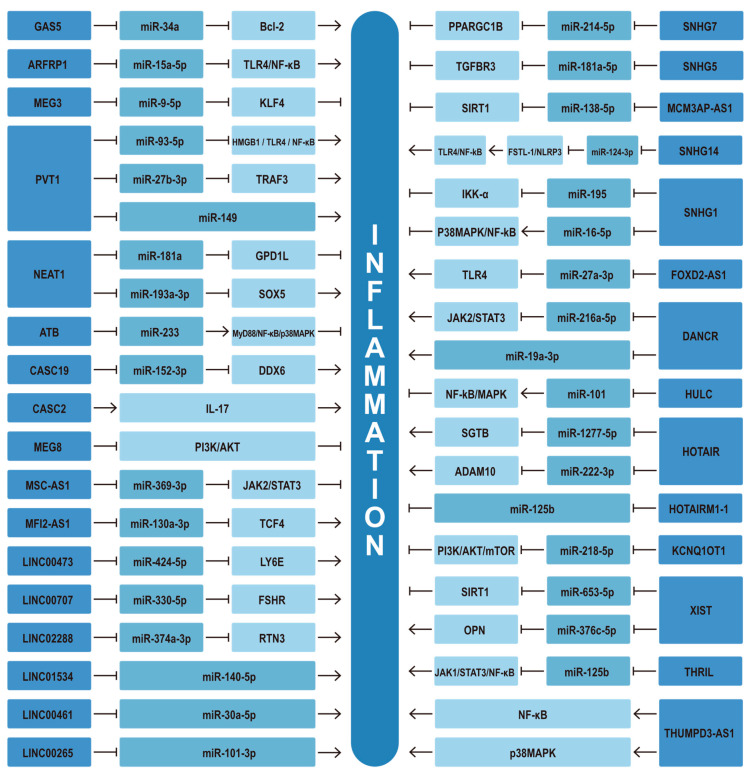
Schematic representation of the signaling pathways involved in this review for the regulation of inflammatory responses by lncRNAs. Different shades of color represent different molecules.

**Figure 4 biomolecules-13-00580-f004:**
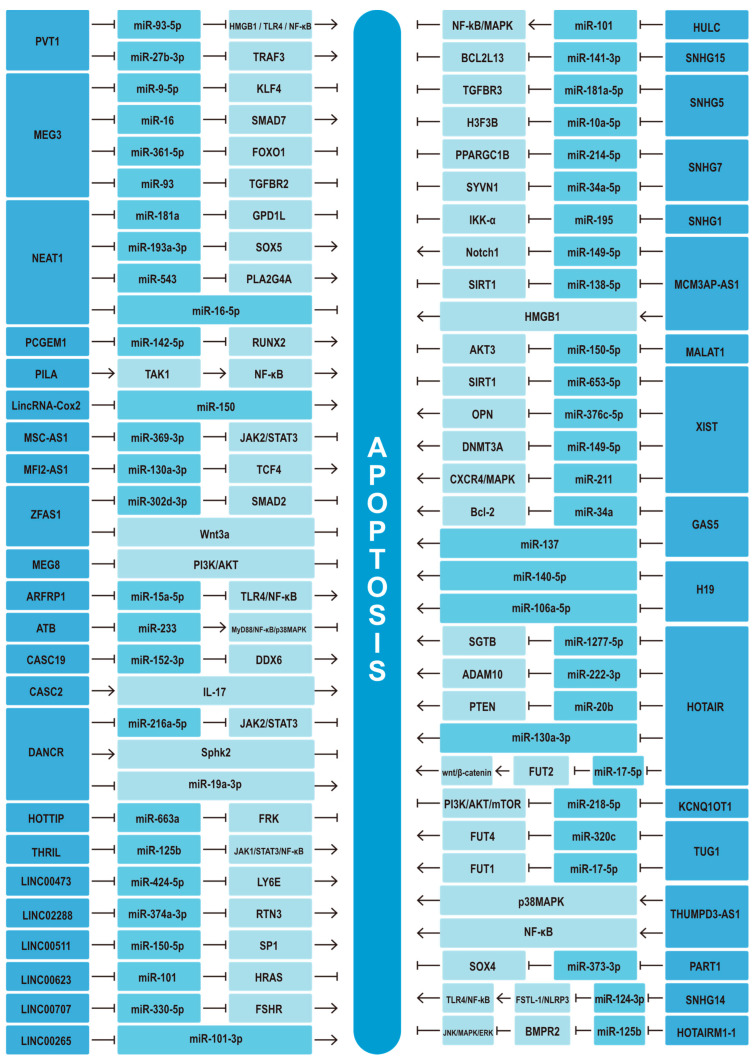
Schematic representation of the signaling pathways involved in this review for the regulation of apoptosis by lncRNAs. Different shades of color represent different molecules.

**Figure 5 biomolecules-13-00580-f005:**
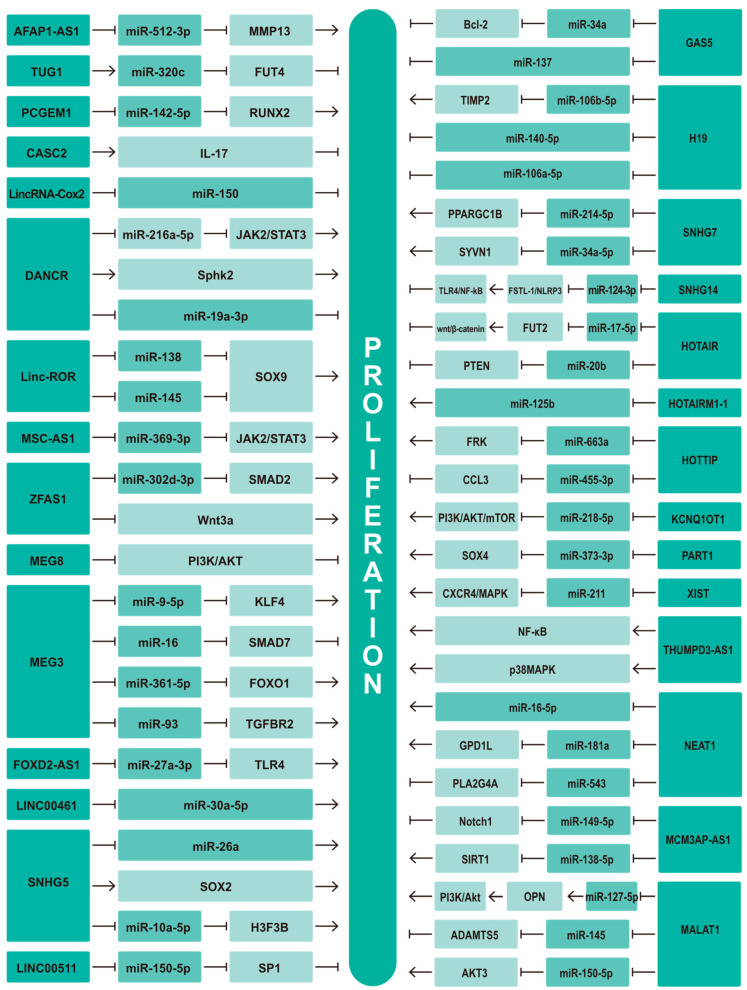
Schematic representation of the signaling pathways involved in this review for the regulation of cell proliferation by lncRNAs. Different shades of color represent different molecules.

**Figure 6 biomolecules-13-00580-f006:**
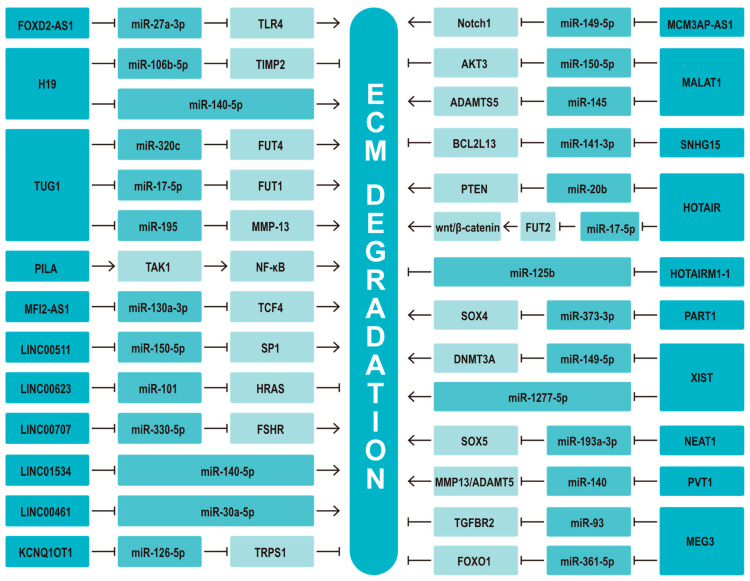
Schematic representation of the signaling pathways involved in this review for the regulation of ECM degradation by lncRNAs. Different shades of color represent different molecules.

**Table 1 biomolecules-13-00580-t001:** Abnormal expression and function of lncRNAs in osteoarthritis cartilage described in this review (animal data).

LncRNA	Expression *	Regulated Axis	Function↑↓	Model
AFAP1-AS1 [124]	up	miR-512-3p/MMP13	proliferation (↑)	In vivo mice models of OA
ATB [50]	down	miR-233	Inflammatory (↓)	The ATDC5 murine chondrocyte cells
			apoptosis (↓)	
H19 [120]	down	miR-106b-5p/TIMP2	proliferation (↑)	Sprague–Dawley rats chondrocytes and FLSs
			ECM degradation (↓)	
HOTAIR [78]	up	miR-17-5p/FUT2	apoptosis (↑)	In vivo rats models of OA
			proliferation (↓)	
			ECM degradation (↑)	
HOTAIR [79]	up	miR-20b/PTEN	apoptosis (↑)	Male adult C57BL/6 mice tissues and chondrocytes
			proliferation (↓)	
			ECM degradation (↑)	
HOTTIP [114]	up	miR-455-3p/CCL3	proliferation (↓)	Male adult C57BL/6 mice tissues
HULC [51]	down	miR-101/NF-kB/MAPK	Inflammatory (↓)	The ATDC5 murine chondrocyte cells
			apoptosis (↓)	
LINC00473 [64]	up	miR-424-5p/LY6E	Inflammatory (↑)	In vivo rats models of OA
			apoptosis (↑)	
			ECM degradation (↑)	
LINC00511 [90]	up	miR-150-5p/SP1	apoptosis (↑)	The ATDC5 murine chondrocyte cells
			proliferation (↓)	
			ECM degradation (↑)	
LINC02288 [66]	up	miR-374a-3p/RTN3	Inflammatory (↑)	In vivo rats models of OA
			apoptosis (↑)	
lincRNA-Cox2 [89]	up	miR-150	apoptosis (↑)	Male adult C57BL/6 mice chondrocytes
			proliferation (↓)	
Linc-ROR [123]	down	miR-138	proliferation (↑)	In vivo nude mice models
		miR-145/SOX9		
MCM3AP-AS1 [97]	up	miR-149-5p/Notch1	apoptosis (↑)	In vivo Sprague–Dawley Rats models of OA
			proliferation (↓)	
			ECM degradation (↑)	
MEG3 [92]	down	miR-9-5p/KLF4	Inflammatory (↓)	The ATDC5 murine chondrocyte cells
			apoptosis (↓)	
			proliferation (↑)	
MEG3 [93]	down	miR-93/TGFBR2	apoptosis (↓)	Sprague–Dawley rats chondrocytes
			proliferation (↑)	
			ECM degradation (↓)	
MEG3 [95]	down	miR-16/SMAD7	apoptosis (↑)	Sprague–Dawley rats chondrocytes
			proliferation (↓)	
NEAT1 [101]	up	miR-16-5p	apoptosis (↓)	The ATDC5 murine chondrocyte cells
			proliferation (↑)	
PILA [91]	up	PRMT1/DHX9/TAK1/NF-κB	apoptosis (↑)	Male adult C57BL/6 mice chondrocytes
			ECM degradation (↑)	
SNHG14 [45]	up	miR-124-3p/FSTL-1/NLRP3/TLR4/NF-kB	Inflammatory (↑)	In vivo rats models of OA
			apoptosis (↑)	
			proliferation (↓)	
SNHG15 [84]	down	miR-141-3p/BCL2L13	apoptosis (↓)	In vivo rats models of OA
			proliferation (↑)	
			ECM degradation (↓)	
SNHG7 [42]	down	miR-214-5p/PPARGC1B	Inflammatory (↓)	Male adult C57BL/6 mice chondrocytes
			apoptosis (↓)	
			proliferation (↑)	
THRIL [60]	up	miR-125b/JAK1/STAT3/NF-κB	Inflammatory (↑)	The ATDC5 murine chondrocyte cells
			apoptosis (↑)	
XIST [119]	up	miR-1277-5p	ECM degradation (↑)	In vivo rats models of OA
XIST [49]	down	miR-653-5p/SIRT1	Inflammatory (↓)	The ATDC5 murine chondrocyte cells
			apoptosis (↓)	
XIST [73]	up	miR-149-5p/DNMT3A	apoptosis (↑)	In vivo Wistar Rats models of OA
			ECM degradation (↑)	

*: Long non-coding RNA expression during osteoarthritis. ↑↓: (↑) means induction, (↓) means inhibition.

**Table 2 biomolecules-13-00580-t002:** Abnormal expression and function of lncRNAs in osteoarthritis cartilage described in this review (human data).

LncRNA	Expression *	Regulated Axis	Function↑↓	Model
AFAP1-AS1 [124]	up	miR-512-3p/MMP13	proliferation (↑)	The human chondrocytes C28/I2 cells
ARFRP1 [58]	up	miR-15a-5p/TLR4/NF-kB	Inflammatory (↑)	Human chondrocytes (isolated from OA cartilage tissues)
			apoptosis (↑)	
CASC19 [57]	up	miR-152-3p/DDX6	Inflammatory (↑)	The human chondrocytes C28/I2 cells
			apoptosis (↑)	
CASC2 [56]	up	IL-17	Inflammatory (↑)	Human chondrocyte cell line CHON-001
			apoptosis (↑)	
			proliferation (↓)	
DANCR [100]	up	miR-577/SphK2	apoptosis (↓)	Human chondrocytes (isolated from OA cartilage tissues)
			proliferation (↑)	
DANCR [37]	up	miR-216a-5p/JAK2/STAT3	Inflammatory (↑)	Human chondrocytes (isolated from OA cartilage tissues)
			apoptosis (↓)	
			proliferation (↑)	
DANCR [38]	up	miR-19a-3p	Inflammatory (↑)	Human chondrocytes (isolated from OA cartilage tissues)
			apoptosis (↑)	
			proliferation (↓)	
FOXD2-AS1 [127]	up	miR-27a-3p/TLR4	Inflammatory (↑)	The human chondrocytes C28/I2 cells
			proliferation (↑)	
			ECM degradation (↑)	
GAS5 [80]	up	miR-34a/Bcl-2	Inflammatory (↑)	Human chondrocytes (isolated from OA cartilage tissues)
			apoptosis (↑)	
			proliferation (↓)	
			ECM degradation (↑)	
GAS5 [81]	up	miR-137	apoptosis (↑)	Human chondrocytes (isolated from OA cartilage tissues)
			proliferation (↓)	
GAS5 [99]	down	miR-146a/Smad4	apoptosis (↓)	Human chondrocytes (isolated from OA cartilage tissues)
H19 [75]	up	miR-106a-5p	apoptosis (↑)	Human chondrocytes (isolated from OA cartilage tissues)
			proliferation (↓)	
H19 [76]	up	miR-140-5p	Inflammatory (↑)	Human chondrocytes (isolated from OA cartilage tissues)
			proliferation (↓)	
			ECM degradation (↑)	
HOTAIR [39]	up	miR-222-3p/ADAM10	Inflammatory (↑)	The human chondrocytes C28/I2 cells
			apoptosis (↑)	
			ECM degradation (↑)	
HOTAIR [40]	up	miR-1277-5p/SGTB	Inflammatory (↑)	Human chondrocyte cell line CHON-001
			apoptosis (↑)	
HOTAIR [77]	up	miR-130a-3p	apoptosis (↑)	Human chondrocytes
HOTAIR [78]	up	miR-17-5p/FUT2	apoptosis (↑)	Human chondrocytes (isolated from OA cartilage tissues)
			proliferation (↓)	
			ECM degradation (↑)	
HOTAIRM1-1 [54]	down	miR-125b	Inflammatory (↓)	Human chondrocytes (isolated from OA cartilage tissues)
			proliferation (↑)	
			ECM degradation (↓)	
HOTAIRM1-1 [55]	down	miR-125b/BMPR2/JNK/MAPK/ERK	apoptosis (↓)	Human mesenchymal stem cells (hMSCs)
HOTTIP [114]	up	miR-455-3p/CCL3	proliferation (↓)	Human mesenchymal stem cells (hMSCs)
HOTTIP [115]	up	miR-663a/FRK	apoptosis (↓)	Human chondrocytes (isolated from OA cartilage tissues)
			proliferation (↑)	
KCNQ1OT1 [112]	down	miR-126-5p/TRPS1	ECM degradation (↓)	Human chondrocyte cell line CHON-001
KCNQ1OT1 [113]	down	miR-218-5p/PIK3C2A/PI3K/AKT/mTOR	Inflammatory (↓)	Human chondrocytes (isolated from OA cartilage tissues)
			apoptosis (↓)	
			proliferation (↑)	
LINC00265 [62]	up	miR-101-3p	Inflammatory (↑)	Human chondrocytes (isolated from OA cartilage tissues)
			apoptosis (↑)	
LINC00461 [63]	up	miR-30a-5p	Inflammatory (↑)	Human chondrocytes (isolated from OA cartilage tissues)
			proliferation (↑)	
			ECM degradation (↑)	
LINC00473 [64]	up	miR-424-5p/LY6E	Inflammatory (↑)	The human chondrocytes C28/I2 cells
			apoptosis (↑)	
			ECM degradation (↑)	
LINC00623 [125]	down	miR-101/HRAS	apoptosis (↓)	Human chondrocytes (isolated from OA cartilage tissues); Human cartilage tissues
			ECM degradation (↓)	
LINC00707 [116]	up	miR-199-3p	apoptosis (↑)	Human chondrocytes (isolated from OA cartilage tissues)
			proliferation (↓)	
LINC00707 [117]	up	miR-330-5p/FSHR	Inflammatory (↑)	Human chondrocytes (isolated from OA cartilage tissues)
			apoptosis (↑)	
			ECM degradation (↑)	
LINC01534 [65]	up	miR-140-5p	Inflammatory (↑)	Human chondrocytes (isolated from OA cartilage tissues)
			ECM degradation (↑)	
LINC02288 [66]	up	miR-374a-3p/RTN3	Inflammatory (↑)	Human chondrocytes (isolated from OA cartilage tissues)
			apoptosis (↑)	
Linc-ROR [123]	down	miR-138	proliferation (↑)	Human bone marrow-derived mesenchymal stem cells (BMSCs)
		miR-145/SOX9		
MALAT1 [109]	up	miR-127-5p	proliferation (↑)	Human chondrocytes (isolated from OA cartilage tissues)
		OPN/PI3K/Akt		
MALAT1 [110]	up	miR-150-5p/AKT3	apoptosis (↓)	Human chondrocytes (isolated from OA cartilage tissues)
			proliferation (↑)	
			ECM degradation (↓)	
MALAT1 [111]	up	miR-145/ADAMTS5	proliferation (↓)	Human chondrocytes (isolated from OA cartilage tissues)
			ECM degradation (↑)	
MCM3AP-AS1 [96]	down	miR-138-5p/SIRT1	Inflammatory (↓)	Human chondrocyte cell line CHON-001
			apoptosis (↓)	
			proliferation (↑)	
MCM3AP-AS1 [97]	up	miR-149-5p/Notch1	apoptosis (↑)	The human chondrocytes C28/I2 cells
			proliferation (↓)	
			ECM degradation (↑)	
MCM3AP-AS1 [98]	up	miR-142-3p/HMGB1	apoptosis (↑)	Human chondrocytes (isolated from OA cartilage tissues)
MEG3 [92]	down	miR-9-5p/KLF4	Inflammatory (↓)	Human chondrocyte cell line CHON-001
			apoptosis (↓)	
			proliferation (↑)	
MEG3 [94]	down	miR-361-5p/FOXO1	apoptosis (↓)	Human chondrocytes (isolated from OA cartilage tissues)
			proliferation (↑)	
			ECM degradation (↓)	
MEG8 [53]	down	PI3K/AKT	Inflammatory (↓)	The human chondrocytes C28/I2 cells
			apoptosis (↓)	
			proliferation (↑)	
MFI2-AS1 [59]	up	miR-130a-3p/TCF4	Inflammatory (↑)	The human chondrocytes C28/I2 cells
			apoptosis (↑)	
			ECM degradation (↑)	
MSC-AS1 [52]	down	miR-369-3p/JAK2/STAT3	Inflammatory (↓)	Human chondrocytes (isolated from OA cartilage tissues)
			apoptosis (↓)	
NEAT1 [102]	up	miR-543/PLA2G4A	apoptosis (↑)	Human chondrocytes (isolated from OA cartilage tissues)
			proliferation (↓)	
NEAT1 [46]	up	miR-193a-3p/SOX5	Inflammatory (↑)	Human chondrocytes (isolated from OA cartilage tissues)
			apoptosis (↑)	
			ECM degradation (↑)	
NEAT1 [47]	down	miR-181a/GPD1L	Inflammatory (↓)	Human chondrocytes (isolated from OA cartilage tissues)
			apoptosis (↓)	
			proliferation (↑)	
PART1 [126]	up	miR-373-3p/SOX4	apoptosis (↓)	Human chondrocytes (isolated from OA cartilage tissues)
			proliferation (↑)	
			ECM degradation (↑)	
PCGEM1 [128]	up	miR-142-5p/RUNX2	apoptosis (↑)	Human chondrocytes and FLSs (isolated from synovium tissues)
			proliferation (↓)	
			ECM degradation (↑)	
PILA [91]	up	PRMT1/DHX9/TAK1/NF-κB	apoptosis (↑)	Human chondrocytes (isolated from OA cartilage tissues)
			ECM degradation (↑)	
PVT1 [118]	up	miR-140/MMP13/ADAMT5	ECM degradation (↑)	Human chondrocytes (isolated from OA cartilage tissues)
PVT1 [34]	up	miR-93-5p/HMGB1/TLR4/NF-κB	Inflammatory (↑)	The human chondrocytes C28/I2 cells
			apoptosis (↑)	
PVT1 [35]	up	miR-27b-3p/TRAF3	Inflammatory (↑)	The human chondrocytes C28/I2 cells
			apoptosis (↑)	
PVT1 [36]	up	miR-149	Inflammatory (↑)	Human chondrocytes (isolated from OA cartilage tissues)
SNHG1 [43]	down	miR-16-5p/P38 MAPK/NF-kB	Inflammatory (↓)	Human chondrocytes
SNHG1 [44]	down	miR-195/IKK-α	Inflammatory (↓)	The human chondrocytes C28/I2 cells
			apoptosis (↓)	
SNHG14 [45]	up	miR-124-3p/FSTL-1/NLRP3/TLR4/NF-kB	Inflammatory (↑)	Human chondrocytes (isolated from OA cartilage tissues)
			apoptosis (↑)	
			proliferation (↓)	
SNHG15 [84]	down	miR-141-3p/BCL2L13	apoptosis (↓)	Human chondrocytes (isolated from OA cartilage tissues)
			proliferation (↑)	
			ECM degradation (↓)	
SNHG5 [121]	down	miR-26a/SOX2	proliferation (↑)	Human chondrocyte cell line CHON-001
SNHG5 [41]	down	miR-181a-5p/TGFBR3	Inflammatory (↓)	The human chondrocytes C20/A4 cells
			apoptosis (↓)	
			ECM degradation (↓)	
SNHG5 [82]	down	miR-10a-5p/H3F3B	apoptosis (↓)	Human chondrocytes (isolated from OA cartilage tissues)
			proliferation (↑)	
SNHG7 [83]	down	miR-34a-5p/SYVN1	apoptosis (↓)	Human chondrocytes (isolated from OA cartilage tissues)
			proliferation (↑)	
THUMPD3-AS1 [61]	down	NF-κB/P38 MAPK	Inflammatory (↑)	The human chondrocytes C28/I2 cells
			apoptosis (↓)	
			proliferation (↑)	
TUG1 [122]	up	miR-195/MMP-13	ECM degradation (↑)	Human chondrocytes (isolated from OA cartilage tissues)
TUG1 [87]	up	miR-320c/FUT4	apoptosis (↑)	The human chondrocytes C28/I2 cells
			proliferation (↓)	
			ECM degradation (↑)	
TUG1 [88]	up	miR-17-5p/FUT1	apoptosis (↑)	Human chondrocytes (isolated from OA cartilage tissues)
			ECM degradation (↑)	
XIST [119]	up	miR-1277-5p	ECM degradation (↑)	Human chondrocytes (isolated from OA cartilage tissues)
XIST [48]	up	miR-376c-5p/OPN	Inflammatory (↑)	Human chondrocytes (isolated from OA cartilage tissues); THP-1 monocytic cell line
			apoptosis (↑)	
XIST [49]	down	miR-653-5p/SIRT1	Inflammatory (↓)	Human chondrocyte cell line CHON-001
			apoptosis (↓)	
XIST [73]	up	miR-149-5p/DNMT3A	apoptosis (↑)	Human chondrocyte cell line CHON-001
			ECM degradation (↑)	
XIST [74]	up	miR-211/CXCR4/MAPK	apoptosis (↑)	Human chondrocytes (isolated from OA cartilage tissues); Human cartilage tissues
			proliferation (↑)	
ZFAS1 [85]	down	miR-302d-3p/SMAD2	apoptosis (↓)	Human chondrocytes (isolated from OA cartilage tissues)
			proliferation (↑)	
ZFAS1 [86]	down	Wnt3a	apoptosis (↓)	Human chondrocytes (isolated from OA cartilage tissues)
			proliferation (↑)	

*: Long non-coding RNA expression during osteoarthritis. ↑↓: (↑) means induction, (↓) means inhibition.

## Data Availability

Not applicable.

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
