# Peer review of "Advances in Research on the Regulatory Roles of lncRNAs in Osteoarthritic Cartilage"

_biomolecules, 2023, doi:10.3390/biom13040580_

Round 1

Reviewer 1 Report (Previous Reviewer 1)

Authors did good revision according to the reviewer's suggestions. All sections and data presented in this manuscript are reseasonable and science significant. This review article provides perspective insign into OA study while it focused on lncRNA, therefore, this article presents valuable information for researchers who are interested in OA therapeutic or prognosis study. 

Author Response

Response to Reviewer 1 Comments

Point: Authors did good revision according to the reviewer's suggestions. All sections and data presented in this manuscript are reasonable and science significant. This review article provides perspective insight into OA study while it focused on lncRNA, therefore, this article presents valuable information for researchers who are interested in OA therapeutic or prognosis study. 

Response: Thank you very much for your kind consideration about our work. We are very grateful for your recognition of this review article.

Reviewer 2 Report (New Reviewer)

This is a fairly comprehensive overview of the expression and functional role of lncRNAs in  OA cartilage.

I have two main recommendations I would like ot see the authors adopt prior to publication.

1. LncRNAs are poorly conserved, particularly at sequence level. Therefore it is important that the authors provide more information in Table 1 on what was the model used. Was it human cartilage/primary chondrocytes/human cell lines. Or rodent cartilage ex vivo tissue/chondrocyte cells or in vivo animal models of OA.  With this additional information Table 1 will become much more informative.  It might also mean that with this extra level of information that the Table needs to be split into a Table 1 (animal data) and a Table 2 (human data).

2. Since OA is now widely recognised as a disease of the whole joint with important cross talk between synovial tissue fibroblasts and chondrocytes, I would like to see the authors provide some acknowledgemnt in the discussion where particular lncRNAs in the cartilage have also been implicated in mediating pathology in the synovium and bone

Minor points

1. The authors should cite the publication by Pearson MJ et al. doi.org/10.1002/art.39520

Identified lncRNAs assocaited with the il-1 inflammatory response in chondrocytes. this included an IL-7 AS lncRNA. 

Author Response

Response to Reviewer 2 Comments

This is a fairly comprehensive overview of the expression and functional role of lncRNAs in OA cartilage.

I have two main recommendations I would like to see the authors adopt prior to publication.

Point 1: LncRNAs are poorly conserved, particularly at sequence level. Therefore it is important that the authors provide more information in Table 1 on what was the model used. Was it human cartilage/primary chondrocytes/human cell lines. Or rodent cartilage ex vivo tissue/chondrocyte cells or in vivo animal models of OA. With this additional information Table 1 will become much more informative. It might also mean that with this extra level of information that the Table needs to be split into a Table 1 (animal data) and a Table 2 (human data).

Response 1: Thank you very much for your constructive comments. We accept your suggestions and have made adequate revisions as your suggested. We have divided the original Table into two parts and added information about the models (Page 15-21, Table 1 and 2).

Point 2: Since OA is now widely recognised as a disease of the whole joint with important cross talk between synovial tissue fibroblasts and chondrocytes, I would like to see the authors provide some acknowledgement in the discussion where particular lncRNAs in the cartilage have also been implicated in mediating pathology in the synovium and bone.

Response 2: Thank you for your valuable suggestions. We accept your suggestions and have carefully added some descriptions of the lncRNAs in the cartilage in the section of in “Summary and Outlook” (Page 22-23, line 538-549), the contributions of these lncRNAs to OA have also been implicated in pathologies of osteoarthritic synovium and subchondral bone.

Minor points

Point 1: The authors should cite the publication by Pearson MJ et al. doi.org/10.1002/art.39520

Identified lncRNAs associated with the il-1 inflammatory response in chondrocytes. this included an IL-7 AS lncRNA. 

Response1 : Thank you for your helpful suggestions. We have accepted your suggestions and have referenced this study in the section of “3.1 Regulation of perichondral inflammation” (Page 7, line 229-233).

This manuscript is a resubmission of an earlier submission. The following is a list of the peer review reports and author responses from that submission.

Round 1

Reviewer 1 Report

1. In this article, it was difficult to understand the regulatory roles of IncRNAs in OA progression without any hierarchy layout in main text. This review article started from “1. Introduction” and then went into the different mechanisms (2. Inflammatory, 3. Apoptosis, 4. Cell proliferation and ECM degradation) directly that regulated by IncRNAs in OA. Please revise main text by following hierarchy layout, 1. Introduction, 2. The introduction of OA pathophysiology (incidence, clinical characteristics, diagnosis, stages, molecular regulatory mechanisms…), 3. The introduction of pathophysiologic role of IncRNAs (genetic regulation, expression, biofunctions, pathological regulation) in OA, 4. Current studies of IncRNAs regulatory roles at different OA progression stage (inflammation, apoptosis, proliferation, ECM degradation…), 5.Current therapeutic findings/clinical trials/ applications that targeting on IncRNAs in OA, 6. Conclusion. If authors had IncRNAs and OA associated publications, they can be described in main text and made a discussion with other studies as well. There are lots of detail needed to be describe with systematic hierarchy layout in this article rather than ambiguous descriptions. Otherwise, the main text in this review article did not present advances and scientific significances. Please make a major revision in main text with hierarchy layout.    

2. In figures, please shows references under all gene names listed in all figures or they were hard to realize the source and what kind of research they did. Or the table 1 can be divided into several parts based on the molecular mechanism (inflammation, apoptosis, proliferation and ECM degradation) and then all figures can be removed from main text.

3. In main text, please rearrange all description based on the scientific significance, importance of findings in OA-related IncRNAs rather than put all findings in it. Otherwise, it was hard to read and understand the advances information from this article. Please revise it.

    4. From line 63 to 67 and references 25, 26 that included in this sentence       did not show significance or association with OA or IncRNAs, especially          references 25, 26 that are talking about cancer. Please clarify this                     paragraph. 

Reviewer 2 Report

In their review Wu and colleagues bring the regulation by long non-coding RNAs into focus as a potential therapeutic target to diagnose and treat osteoarthritis, a common degenerative joint disorder with limited treatment options. The authors summarize and give a comprehensive overview of recently published research on lncRNAs in the context of OA. Nevertheless, the manuscript is written in a narrative manner and mainly lists the findings of various studies (either supporting or opposing each other) and is therefore rather descriptive, not conclusive. In my opinion, the review as it is now does not identify a clear authors’ point of view to the given topic, except that it is “a hot topic” as the authors have said. Nor does it fully meet the aim “to further elucidate the pathogenesis of OA and provide insights for the diagnosis and treatment of OA” set by the authors in the abstract and introduction. I think the authors should expand substantially the Introduction and Summary/Outlook parts by discussing these aims more specifically and/or add a part dedicated to the mechanism, diagnosis and treatment of OA with regard to lncRNAs.

To be more specific, I would like to break it down to the following key points:

1). The authors poorly – if at all – introduce lncRNAs as biologically active molecules at the beginning of the review. Introduction on how lncRNAs regulate gene expression and how they are regulated, supported by illustrations, will be helpful.

2). Table 1 summarizes the content of the review sufficiently enough and thus somewhat questions the value of the rest of the manuscript and of the figures. Meanwhile, the functions and therapeutic potential of lncRNA listed in the table/text have barely been mentioned or discussed, which unfortunately makes the manuscript rather a list of facts.

3). The actual goal and the outlook of this review remain unfortunately unclear. Which of the lncRNAs mentioned and what pathways/mechanisms of their regulation do the authors consider as the most promising therapeutic or diagnostic targets in OA and why? Are there any challenges in the lncRNA-based diagnostics and therapy of OA? Please discuss in the summary or in a separate section. Besides, in the outlook the authors say “Therefore, it is possible to ameliorate or even reverse the progression of OA by regulating these differentially expressed lncRNAs” – please discuss how exactly this could be accomplished.

Round 2

Reviewer 1 Report

Major comments:

1.      In figure 1, please mark 5 types of IncRNAs with numbers of (1), (2),….(5). It would be better if (3) bidirectional, (4) intronic, or (5) intergenic can be showed separately rather than showed in the same sequences. Please moves the upper panel to a little left, it was out of edge of the paper. Please indicates what’s the purposes or indications of arrows with different color. The fragment size of intro always bigger than exon, and are IncRNA fragment sizes more bigger than intro and exon? Please clarify them in this figure. Please cites the references before “(Fig. 1)” of this paragraph, or it is hard to know that the 5 types of IncRNAs were identified by authors or another researchers.

2.      In line 111, since “3.1 Regulation of perichondral inflammation” as a sub-title, the paragraph of “Currently, the inflammatory….” would be started from line of 112. Please clarify it. The same problems are presented in sub-title of 3.2 and 3.3 as well.

3.      Please cite the references for the paragraph from line 120 to line 123.

4.      In figure 6, it was divided into upper panel and down panel, however, which presented no differences between two panels except color and citations. Please identify or indicate what are the different between these two panels in figure 6.

5.      Please unify the words’ size of each figure legend and make the size different from main text, it would be easy to understand the different paragraph within the main text. And paragraph is not necessary in the figure 2 legend.

Reviewer 2 Report

1. The title of a newly added sub-chapter “Diagnosis and treatment of lncRNA in OA” is senseless. Besides, in this sub-chapter the authors provide examples of lncRNA as biomarkers of OA but do not give a comprehensive overview of lncRNA-based or lncRNA-targeted therapeutic options. Instead, they exclusively discuss plant-derived preparations for the treatment of OA. If the authors consider plant extracts as the most promising treatment in OA compared to other lncRNA-based therapeutic strategies, this should be justified clearly in this sub-chapter or in the summary.

2. The amendments made by the authors to “Summary and Outlook” are unfortunately misleading, for example:

Almost all of the current studies on lncRNAs are preclinical, we consider that one major reason for this may be owing to the lack of effective intraarticular delivery systems”. Is intraarticular injection not effective enough in authors’ opinion? Why?

New delivery strategies have been developed recently to reduce off-target effects, such as nanotechnology-based drug delivery systems and exosomes with target recognition. Based on this, we believe that it is feasible to include lncRNA molecule as a diagnostic biomarker or inhibit the development of OA by local injection”. By local injection of lncRNA or its effector? What do the authors mean by “local”? Why do the authors prefer “local injection” to the other delivery strategies mentioned above?

3. The layout of the newly added Fig. 1 is very misleading and even delivers wrong information (see “sense/antisense”, “bidirectional”)

4. More references should be provided to support the mechanisms depicted in Fig. 2.

5. Lines 82-91: The sentence is hard to read and follow.